# Epidemiology of Viral Hepatitis in the Indigenous Populations of the Arctic Zone of the Republic of Sakha (Yakutia)

**DOI:** 10.3390/microorganisms12030464

**Published:** 2024-02-25

**Authors:** Vera S. Kichatova, Maria A. Lopatukhina, Ilya A. Potemkin, Fedor A. Asadi Mobarkhan, Olga V. Isaeva, Mikhail D. Chanyshev, Albina G. Glushenko, Kamil F. Khafizov, Tatyana D. Rumyantseva, Sergey I. Semenov, Karen K. Kyuregyan, Vasiliy G. Akimkin, Mikhail I. Mikhailov

**Affiliations:** 1Central Research Institute of Epidemiology, 111123 Moscow, Russia; marialopatukhina@yandex.ru (M.A.L.); axi0ma@mail.ru (I.A.P.); 1amfa@bk.ru (F.A.A.M.); isaeva.06@mail.ru (O.V.I.); chanyshev@cmd.su (M.D.C.); glushchenko.a@cmd.su (A.G.G.); khafizov@cmd.su (K.F.K.); karen-kyuregyan@yandex.ru (K.K.K.); vgakimkin@yandex.ru (V.G.A.); michmich2@yandex.ru (M.I.M.); 2Mechnikov Research Institute of Vaccines and Sera, 105064 Moscow, Russia; 3Russian Medical Academy of Continuing Professional Education, 125993 Moscow, Russia; 4Science and Research Department, Arctic State Agrotechnological University, 677008 Yakutsk, Russia; tanya_rum@mail.ru; 5Research Center, Ammosov North-Eastern Federal University, 677010 Yakutsk, Russia; insemenov@yandex.ru

**Keywords:** Hepatitis A virus, Hepatitis E virus, Hepatitis B virus, Hepatitis delta virus, Hepatitis C virus, genotypes, seroprevalence, Arctic, indigenous people, public health

## Abstract

The indigenous populations of the Arctic regions of Russia experience the lowest coverage of health-related services. We assessed the prevalence of hepatitis A, B, C, D and E viruses (HAV, HBV, HCV, HDV and HEV) among 367 healthy adult Native people of the Arctic zone of Yakutia. The HAV seroprevalence was above and increased with age. The anti-HEV IgM and IgG antibody detection rates were 4.1% and 2.5%, respectively. The average HBsAg detection rate was 4.6%, with no positive cases identified in participants aged under 30 years, confirming the effectiveness of the newborn vaccination program that began in 1998. Anti-HDV antibodies were detected in 29.4% of HBsAg-positive cases. The anti-HCV and HCV RNA detection rates peaked in the age cohort of 50–59 years (10.8% and 3.9%). No statistically significant gender differences in the prevalence of different viral hepatitis were observed. The time-scaled phylogenetic analysis demonstrated that all HBV genotype A and D strains isolated in this study were autochthonous and had an estimated most common recent ancestor (MCRA) age of around the 11th to 14th century. Unlike HBV, the HCV strains of subtypes 1b, 2a and 2k/1b were introduced from other regions of Russia in the 1980s and 1990s. The HCV 1b sequence analysis revealed a series of transmission events. In conclusion, these data emphasize the urgent need for expanded viral hepatitis screening and care programs in the indigenous populations of the Arctic zone of Yakutia.

## 1. Introduction

Due to its high prevalence, blood-borne viral hepatitis, mainly types B and C, poses a significant threat to the health of the indigenous populations of the Circumpolar Arctic; however, the hepatitis B burden is gradually decreasing due to the infant vaccination program [1,2,3]. These hard-to-reach and significantly affected populations are one of the most important groups that require targeted hepatitis prevention, testing and treatment services to achieve the key indicators of the World Health Organization (WHO) viral hepatitis elimination program, defined as a 90% decrease in incidence and a 65% decrease in mortality [4]. 

The burden of enteric viral hepatitis, namely hepatitis A and E, in the indigenous populations of the Circumpolar Arctic, has been less studied. Hepatitis A virus (HAV) infection that is transmitted through the fecal–oral route was previously reported to be prevalent in the Arctic indigenous populations based on seroprevalence studies [5]; however, HAV has now become rare in some populations, such as the Alaska Native persons, due to childhood vaccination programs [6]. However, the current data on HAV seroprevalence are limited in relation to many groups within the population of the Circumpolar Arctic. 

Data on hepatitis E virus (HEV) in the Circumpolar Arctic are scarce. Although no reports of acute hepatitis E in the Arctic have been published so far, the data on the detection of antibodies to HEV (anti-HEV) in Alaska Native peoples, Canadian Inuits and Yakutian reindeer herders [7,8,9] suggest the possible relevance of this infection, often zoonotic in temperate territories [10], for the indigenous populations of the Circumpolar Arctic.

The Republic of Sakha (Yakutia) is a region of the Russian Federation with high incidence rates of chronic viral hepatitis B and C that significantly exceeded the Russian average until 2020, i.e., before the COVID-19 pandemic that significantly affected incidence reporting (31.7 and 45.9 per 100,000 for HBV and HCV, respectively, in Yakutia vs. 11.4 and 37.1 per 100,000 national ten-year average), and high mortality levels due to liver cirrhosis and hepatocellular carcinoma (HCC) [11,12,13]. A significant proportion of this territory is located north of the Arctic Circle (latitude 66° N). The administrative districts of Yakutia differ significantly in the size and ethnicity of their populations, as well as in their inaccessibility and the degree of climatic severity. The remoteness from the regional centers of the republic, together with either absent or poor transport accessibility, as well as low population density (0.32 people/km^2^ [14]) critically and negatively influence the standard of living of the local population, including the availability of the universal and specialized medical care compared to other regions of Russia and migrant population within the region, which is concentrated mainly in cities. This problem is most acute in the Arctic regions of Yakutia, where the population density does not exceed 0.08 people/km^2^ [15], and the population is largely composed of the indigenous people of the North—Evenks, Evens and Yukaghirs [16]. The prevalence of the hepatitis B virus (HBV), hepatitis D virus (HDV) and hepatitis C virus (HCV) in these populations varies significantly depending on the studied region and/or ethnic group [17]. In many native populations of Northern Siberia, the HBV infection is considered to be endemic based on the high detection rate of viral antigens and antibodies and the autochthonous HBV sub-genotypes observed in these groups [17,18]. However, due to the low coverage of the indigenous people of Yakutia with viral hepatitis screening programs and other related services, there are significant gaps in knowledge about the burden of hepatitis infections and genetic variants in the viruses circulating in these native populations. The objective of this study was to assess the prevalence of hepatitis A, B, C, D and E viruses and/or herd immunity to these infections among the Native people of the Arctic zone of Yakutia to determine the genetic diversity of HBV and HCV, and to reconstruct the history of the introduction of these two viruses into the studied population using Bayesian analysis.

## 2. Materials and Methods

### 2.1. Serum Samples

Serum samples from 367 indigenous inhabitants of the Arctic zone living in six settlements located in the Momsky district of Yakutia were included in the current study. The sera collection was carried out as part of a regional project intended to assess the quality of life and health status of the local population. All district inhabitants were invited for medical check-ups via radio, local television and by notifying heads of local communities. Viral hepatitis testing was offered to all people who came for a check-up. The number of study participants in each settlement is shown in Figure 1, together with the settlement’s location and the total population. The proportion of people surveyed was 9.0% of the total population living in these six settlements, or 8.2% (367/4452) of the total population of the Momsky district of Yakutia, based on reported statistical data [19]. Demographic data (gender, age, nationality, place of residence) were collected from each participant using a questionnaire. The inclusion criteria were the written informed consent to participate in the survey and the permanent residence in the study region. Knowledge of the participant’s infection status was not an exclusion criterion, and this question was not addressed during sampling. However, available medical records of participants who tested positive in the current study were checked afterward to assess their knowledge of their infection status. Blood sampling was performed in August 2022. Serum separation was performed by natural sedimentation for 24 h at 18–20 °C. Serum samples were transferred into sterile polypropylene tubes and stored at 2–8 °C for no more than 2 days; then, samples were frozen at −18 to −20 °C and transported, maintaining a cold chain, to the laboratory, where they were stored at −70 °C until testing.

### 2.2. Viral Hepatitis Testing

All serum samples were tested for IgG antibodies to HAV (anti-HAV IgG), HBV surface antigen (HBsAg) and antibodies to HBV core antigen (anti-HBc) using commercially available enzyme-linked immunosorbent assay (ELISA) kits (Vector-Best, Novosibirsk, Russia), as well as anti-HEV IgG and IgM antibodies and total antibodies to HCV (anti-HCV) using commercially available kits (Diagnostic Systems, Nizhniy Novgorod, Russia). All HbsAg-positive sera were tested for antibodies to HDV (anti-HDV) using a commercially available ELISA kit (Vector-Best, Novosibirsk, Russia) and HBV DNA using a commercially available assay with the limit of detection 50 IU/mL (AmpliSens^®^HBV-FL kit; AmpliSense, Moscow, Russia). Anti-HDV reactive sera were tested for HDV RNA using the reverse transcription polymerase chain reaction (RT-PCR) protocol with primers to HDV R0 fragment described elsewhere [20]. All anti-HCV reactive samples were tested for HCV RNA using an AmpliSens^®^HCV-FL kit (AmpliSense, Moscow, Russia). A brief description and key performance characteristics, such as specificity, analytical sensitivity and diagnostic sensitivity for all commercial serological assays used in the study, are given in Appendix A. 

Sera reactive for anti-HEV IgM were tested for HEV RNA using RT-PCR with degenerate nested primers targeting the open reading frame 2 (ORF2) region [21].

All testing performed with commercial ELISA and PCR assays was completed according to the instructions of the manufacturers of the respective kits. PCR testing was performed immediately after the extraction of nucleic acids.

### 2.3. HBV and HCV Amplification, Sequencing and Genotyping

Viral nucleic acid extraction was performed using the QIAamp Viral RNA Mini Kit (QIAGEN, Hilden, Germany). For the amplification of the entire HBV genome, a panel of primers containing Illumina adapter tails was employed. Multiplex PCR amplification was run in two separate reactions containing 10 μL of DNA, 10 μL of PCR-mix-2-blue (AmpliSense, Moscow, Russia), 1.4 μL of dNTP 4.4 mM (AmpliSense, Moscow, Russia), the primers (the final concentration of each primer in the reaction mixture and pool number are presented in Appendix A) and sterile water in a final volume of 25 μL. The amplification profile was as follows: (1) denaturation at 95 °C for 3 min; (2) 16 amplification cycles, 95 °C—30 s, 55 °C—30 s and 72°C—20 s; (3) final elongation at 72 °C for 3 min. Pooled PCR products were purified using AMPure XP beads (Beckman Coulter, Indianapolis, IN, USA) at a ratio of 1:1; the elution volume was 15 μL. Indexing PCR was performed with 10 μL of PCR-mix-2-blue (AmpliSense, Moscow, Russia), 1.4 μL of dNTP 4.4 mM (AmpliSense, Moscow, Russia), 5 μL of purified PCR products, sterile water and Nextera index adapters (Appendix A), with final concentration of each primer 200 nM and final reaction volume 25 μL. The amplification profile was as follows: (1) denaturation at 95 °C for 1 min; (2) 25 amplification cycles, 95 °C—20 s, 55 °C—30 s and 72°C—20 s; (3) final elongation at 72 °C for 3 min. Pooled products of indexing PCR were purified using AMPure XP beads (Beckman Coulter, Indianapolis, IN, USA) at a ratio of 1:1. Concentration measurement for purified libraries was performed with Qubit dsDNA HS Assay Kit on Qubit 4.0 fluorimeter (Invitrogen, Waltham, MA, USA). High-throughput sequencing was performed on the Illumina MiSeq platform with MiSeq Reagent Kit v2 (300 cycles) according to the manufacturer’s instructions.

In the samples positive for HBsAg but negative for viral DNA, the HBV genotype was predicted based on HBsAg serotyping performed using the ELISA kit (Vector-Best, Novosibirsk, Russia). In this assay, three different monoclonal antibody-based conjugates were used in parallel for HBsAg detection was performed in parallel allowing differentiation between the following HBV serotypes/genotypes: ayw2/D or ayw3/D, adw2/A, adrq+/C.

The 942 nt long fragment of the HCV genome encoding core and E1 proteins was amplified from all HCV RNA-positive samples (nucleotide positions 293–1234 according to the reference sequence of HCV 1a strain H77; GenBank accession number AF011753), using the protocol described previously [22]. To confirm the recombinant nature of the HCV sequence in one sample with a 2k subtype identified based on analysis of core fragments, the partial NS5B sequence was amplified using the primers described elsewhere [23]. The HCV amplification products were purified from agarose gel using the QIAquick Gel Extraction kit (QIAGEN, Hilden, Germany) and subjected to Sanger sequencing. 

### 2.4. Time-Scaled Phylogenetic Analysis

To determine HBV and HCV genotypes and establish the genetic relatedness of the identified viral sequences, a time-scaled phylogenetic analysis was performed. 

HBV and HCV datasets were aligned in the MEGA 7 software using the ClustalW algorithm [24]. Bayesian analysis was run in the BEAST v1.10.4 software package. Due to the limited number of HBV complete genome sequences from different parts of Russia and neighboring countries, the time-scaled phylogenetic analysis was restricted to the 676 nt region of the S-gene (nucleotide positions 149–824 by reference sequence NC_003977.2) that is most represented in the GenBank database. The HBV sequences from this study were supplemented with 522 sequences from GenBank, including those recommended by ICTV classification for HBV genotype and subgenotype designation [25], ancient HBV sequences extracted from mummies and archival 59 HBV sequences isolated in different regions of the Russian Federation. For the HBV sequence dataset, run parameters were the same as described previously [26]. 

For HCV Core/E1 fragment sequences, the dataset was supplemented with HCV subtype reference sequences according to the ICTV recommendations from March 2022 [27], all available archival sequences isolated within the territory of the Russian Federation, as well as 10 sequences from the GenBank per sample under study, which showed the maximum degree of identity. All duplicate reference sequences were removed from the final dataset. For all sequences, the year and country of isolation were known, as well as the city/region in the case of sequences from the Russian Federation. The run parameters were as follows: “BEAST Model Test”, relaxed log-normal clock with initial clock rate 5.58 × 10^−4^ substitutions/site/year and population model–constant (effective) population size. The Markov chain Monte Carlo (MCMC) method was run for 50 million generations and sampled every 5000 steps in two repetitions. Tracer v1.6 software was used for convergence assessment. The effective sample size (ESS) was >200. 

The presence of a correlation between genetic divergence and sampling time in HBV and HCV datasets was confirmed by running the TempEst v.1.5 software. The linear regression curves were observed for both HBV and HCV datasets (Appendix A). The FigTree v.1.4.3 software was used for the visualization of phylogenetic trees. 

### 2.5. Statistical Analysis

Statistical analysis was performed using GraphPad 10.0.2 software (https://www.graphpad.com/ (accessed on 20 January 2023)) and included the calculation of the mean values, the 95% confidence interval (95% CI) and assessment of the differences between study groups using Fisher’s exact test for categorical data and Student’s *t*-test for continuous data (significance threshold *p* < 0.05).

## 3. Results

### 3.1. Study Population

The male-to-female ratio in the studied cohort was 1:1.8. The study included individuals from 16 to 92 years of age (median age 55 years). The participants’ median age and gender ratio were similar in all six surveyed settlements (Figure 1). According to the questionnaire data, the majority of participants self-identified as Yakuts (49.6%), Evens (38.7%), Evenks (10.1%), Dogans (0.5%), Russians (0.5%) and 0.5% of participants did not answer this question.

### 3.2. HAV and HEV Testing Results

The average anti-HAV IgG antibody detection rate was 79.6% (292/367, 95% CI: 75.1–83.4%). When comparing different age groups in the studied population, a statistically significant increase in anti-HAV positivity rates in individuals above 40 years was observed (*p* = 0.0140, as shown in Figure 2). The anti-HAV IgG detection rates were similar between male and female participants, 84.4% and 76.7%, respectively (*p* = 0.0823). The analysis of anti-HAV detection rates in participants from the three settlements with the largest sample size (settlements #2 (n = 205), #4 (n = 54) and #6 (n = 65), as indicated in Figure 1) demonstrated the highest seropositivity rate of 94.4% (51/54; 95% CI: 84.3–98.7%) in one particular settlement (#2) that significantly exceeded the observed average level of 79.6% (*p* = 0.0077).

The average anti-HEV IgM and IgG detection rates were 4.1% (15/367, 95% CI: 2.4–6.7%) and 2.5% (9/367, 95% CI: 1.2–4.7%), respectively. The age-specific anti-HEV IgM and IgG rates are shown in Figure 3. No significant differences were observed in anti-HEV IgM and IgG positivity rates between settlements or depending on the gender of study participants.

The anti-HEV IgG detection rates among different age cohorts were similar, while anti-HEV IgM antibody detection rates peaked among people aged between 16 and 29 years old (12.8%; 5/39; 95% CI: 5.1–27.1%) and significantly exceeded the average rate of 4.1% in the surveyed population (*p* = 0.0332). Among 15 anti-HEV IgM reactive samples, anti-HEV IgG antibodies were detected in only 1 sample (6.7%, 1/15). The mean cut-off index (COI; signal sample/cut-off) in anti-HEV IgM reactive samples was 1.95 in those samples negative for anti-HEV IgG and 1.5 in those samples reactive for anti-HEV IgG. Out of 14 anti-HEV IgM reactive/anti-HEV IgG nonreactive samples, 9 (64%) had a COI ≤ 1.5. HEV RNA was not detected in any of the anti-HEV IgM reactive samples.

### 3.3. HBV and HDV Testing Results

HBsAg was detected in the sera of 4.6% (17/367; 95% CI: 2.9–7.3%) of this study’s participants. Six (35.3%) of the seventeen HBsAg-positive participants tested positive in the past, based on available medical records, and, thus, were aware of their infection status. None of them received antiviral treatment. No single HbsAg-positive case was identified among participants under 30 years old, while HBsAg positivity rates peaked in the cohort aged between 40 and 49 years old (Figure 4A). However, the differences in HBsAg positivity rates were not statistically significant between age cohorts and between male and female participants. 

The average anti-HBc positivity rate in the surveyed population reached 59.4% and was similar among male and female participants (60.0% (81/135) vs. 59.1% (137/232), *p* = 0.9124). The anti-HBc detection rate was as low as 15.4% (6/33, 95% CI: 6.9–30.1%) in participants aged less than 30 years but increased significantly in older age groups and reached a plateau at above 60% in the participants aged 40 years and older (Figure 4B).

Among 17 HBsAg-positive participants, anti-HDV antibodies were detected in 29.4% (5/17; CI: 13.0–53.4%), and all reactive sera were obtained from individuals aged 40 years and older (Figure 4A). However, the mean age of anti-HDV-positive and anti-HDV-negative participants reactive for HBsAg did not differ significantly (52.4 years vs. 55.8 years, *p* = 0.6035, Student’s *t*-test). Anti-HDV-positive individuals lived in three of the six study settlements: Sasyr (n = 1), Khonuu (n = 2), Kulun-Elbiut (n = 2). No statistically significant differences in anti-HDV detection rates were observed depending on the gender or place of residence of the participants (*p* = 0.6000, Fisher’s exact test). HDV RNA was detected in three out of five anti-HDV-positive samples. However, due to the small amplicon yield in PCR and the limited volume of original serum samples, these HDV RNA-positive cases were not sequenced.

Of the 17 HbsAg-positive samples, HBV DNA was detected in 9 (53%) samples. The HBV genotype was determined for 15 out of 17 HBsAg-positive sera, based on the combined data from the phylogenetic analysis of the nucleotide sequences from 9 HBV DNA-positive samples and serotyping ELISA assay for 6 HbsAg-positive/HBV DNA-negative samples. Two HbsAg-positive/HBV DNA-negative samples were unavailable for HBV serotyping due to the insufficient volume of the sample. The distribution of HBV genotypes in the studied cohort is shown in Figure 5. Genotypes A (sub-genotype A2) and D (sub-genotypes D1, D2 and D3) were identified, with the latter being slightly more prevalent (genotype D–60%, genotype A–40%). HBV DNA was undetectable in five samples reactive for anti-HDV antibodies. Based on HBV genotype prediction using ELISA assay, genotype D was identified in three anti-HDV-positive samples and genotype A—in one sample. The fifth anti-HDV reactive sample was not available for testing due to insufficient volume.

### 3.4. HCV Testing Results

The anti-HCV detection rate in the surveyed population was 5.2% (19/367; 95% CI: 3.3–9.1%). The anti-HCV detection rates in men and women were identical—5.2% (7/135 and 12/232, respectively, *p* = 1.0000). The cases positive for anti-HCV were detected in all cohorts aged over 30 years, with peak detection rates observed in the cohort aged between 50 and 59 years (Figure 6). HCV RNA was detected in 36.8% (7/19; 95% CI: 19.1–59.1%) of the seropositive participants. Overall, the prevalence of active HCV infection among the surveyed population was 1.9% (7/367; CI: 0.9–4.9%). All cases of active HCV infection were detected in participants aged 40 years and older (Figure 6), with no significant differences depending on the gender or living in a particular settlement. Four (57.1%) of seven HCV RNA-positive participants were aware of their infection status, as they had tested positive in the past. None of them received hepatitis C therapy, based on data from medical records.

### 3.5. HBV and HCV Time-Scaled Phylogenetic Analysis

The HBV and HCV sequences from this study were deposited in GenBank under accession numbers OR947658–OR947666 and OR947667–OR947673, respectively. The time-scaled phylogenetic analysis demonstrated that the sequences of HBV genotype A2 isolated from indigenous inhabitants of the Arctic zone formed two separate clusters with other sequences of this genotype isolated previously in Yakutia (Figure 7). The separation of these genotype A2 clusters from the most recent common ancestor (MRCA) is estimated to have occurred around the 13th–14th centuries, indicating the autochthonous nature of these sequences isolated from the indigenous populations of the Arctic zone. Likewise, the MRCA age estimates for sub-genotype D1, D2 and D3 sequences isolated in this study were from the 11th to 14th centuries, respectively (Figure 7).

Bayesian phylogenetic analysis was performed to identify HCV genotypes and to assess the relatedness of identified sequences. The resulting phylogenetic tree is shown in Figure 8. 

The results of the analysis showed that one HCV sequence from this study belonged to genotype 2a and was grouped together with other sequences of this genotype of Russian origin; another sequence belonged to the recombinant form 2k/1b (confirmed by sequencing of NS5B fragment) and was grouped with 2k/1b sequences previously isolated in the Moscow region (Figure 8). The estimated MRCA time for these two sequences has been dated back to the 1980s to 1990s, indicating that the introduction of these strains from different parts of Russia to the indigenous populations of the Arctic zone occurred around this time. The remaining five HCV sequences from this study were isolated from residents of one particular settlement (#2) and belonged to subtype 1b. These sequences formed a monophyletic group with the MRCA time dated back to 1998 (95% highest posterior density (HPD): 1989–2003) and were close to HCV 1b sequences that were mostly of Russian Far Eastern origin (Figure 8). The topology of the tree branch suggests that sequential HCV transmission events took place among residents of the settlement in the period between 1999 and 2009.

## 4. Discussion

The indigenous populations of the remote Arctic regions of the Russian Federation experience the lowest provision of health-related services, including viral hepatitis screening and ancillary care programs [28]. The largest Arctic region in the Russian Federation is Yakutia, with 79.2% of its territory belonging to reindeer herding farms and physically represented by tundra, forest–tundra, mountain taiga and taiga natural zones [29]. As of 2021, there were only five outpatient clinics and one hospital in Momsky district [14]. However, there has been a trend in recent years towards improving the availability of medical care, including mobile medical teams, air ambulances, telemedicine and increasing the number of medical personnel. Also, the regional legislation of the Arctic zone of the Russian Federation defines the right to preferential or free drug provision for indigenous people living in rural Arctic areas [30]. No hepatitis screening and treatment programs targeting the general population are currently implemented in the region, except HBV and HCV screening of blood donors, hospitalized patients before surgery and testing of household contacts of hepatitis patients. In our study, we addressed the gaps in the current knowledge of the viral hepatitis burden and circulation patterns of hepatitis viruses in indigenous populations in one particular region of the Arctic zone of Yakutia.

The study was conducted in Momsky district, whose population is characterized by socio-demographic attributes typical of the Arctic zone of Yakutia. All residents of the Momsky district are represented by the rural population. The average age of the population is 30 years, with men to women ratio being about 1:1.1. The majority of the population are Yakuts (67.0%), Evens (11.8%), Evenks (0.5%) and Yukaghirs (0.1%). As for 2021, the population decline was 1.0 per 1000 persons [14].

The HAV seroprevalence above 50% in individuals aged over 15 years old in the studied cohort suggests an intermediate level of endemicity, according to the WHO classification [31]. Age-specific HAV seroprevalence rates observed in this study are similar to those obtained in the 2020 serosurvey that primarily targeted the urban population of Yakutia [32]. The significantly lower anti-HAV IgG detection rates in individuals aged under 40 years old compared to older age groups suggest the possible changes in HAV endemicity over the last few decades. However, since only adult participants were surveyed in our current study, the exact age-specific changes in the susceptibility to new HAV infections in indigenous populations remain unclear. Thus, further studies of HAV seroprevalence in children are needed to understand the possible transition in endemicity level in indigenous populations and the necessity of childhood vaccination programs. 

The main reservoirs of HEV infection for humans in temperate regions are domestic pigs and wild boars. Due to severe climatic conditions, the Arctic zone of Yakutia is not a habitat for wild boars. Moreover, no pig farming has existed in this region since the 1990s; in fact, only 22 domestic pigs were registered in the region under the current study (Momsky district) in 2020 [33]. The anti-HEV IgG detection rate in the studied cohort (2.5%) is similar to those observed previously in a small cohort of reindeer herders from another region of Yakutia (4.7%, 4/86) [9] but is substantially lower compared to the data reported for the adult population of the non-Arctic zone of Yakutia (≥6.6%) [34]. The anti-HEV IgG detection in the studied cohort is unlikely to be related to non-specific reactivity, as the ELISA assay used in this study has been shown to be highly specific [35]. Similar anti-HEV IgG positivity rates were reported in cohorts of Canadian Inuits and Alaska Native persons [7,8], confirming HEV circulation and infection in humans in regions where the swine reservoir of the virus is limited or even absent. Reports on anti-HEV detection in reindeer [9,36,37,38], together with the fact that this deer species is a main source of meat for Arctic indigenous populations, suggest that reindeer could be a possible source of zoonotic infection with HEV or another antigenically similar virus, as different hepeviruses exhibit the broad cross-reactivity in serological tests [39]. The low HEV seroprevalence in the population of the Artic zone indicates a low risk of HEV infection. However, the relatively high proportion of sera reactive for anti-HEV IgM in our study (4.1%) is suspicious, given the fact that all study participants either had no symptoms of acute illness or had not reported a recent sickness. Furthermore, only one out of fifteen reactive sera also contained IgG antibodies to HEV. Nevertheless, the anti-HEV IgM assay used in this study has been shown to have a sensitivity and specificity of 98% (95% CI, 88–99.9%) and 95.2% (95% CI, 91.3–97.4%), respectively [40], and the majority of anti-HEV IgM reactive samples had COI values of 1.5 or higher, suggesting the low possibility of false-positive results. The failure to detect HEV RNA in sera reactive for anti-HEV IgM is also non-indicative of non-specific reactivity, as viremia during HEV infection can be transient [41]. Taken together, these data provide evidence of the possible recent asymptomatic HEV infection in those participants reactive for anti-HEV IgM.

Our data indicate that the prevalence of HBV, HDV and HCV is still high among the indigenous populations of the Arctic zone of Yakutia. The analysis of the published data suggests that the last study on the prevalence of these infections in the Arctic regions of Yakutia was performed about 20 years ago [16]. A comparison of the results of the current study with those obtained 20 years ago demonstrates a clear trend toward a decline in HBV prevalence. In the early 2000s, the HBsAg detection rate in individuals aged 15–19 years old was 21.6% [16]. In the current study, no single case of HBsAg detection was identified among participants aged less than 30 years old. The overall HBsAg prevalence also decreased significantly, from 8.0% in the early 2000s [16] to 4.6% in the current study (*p* = 0.0188). Such a decrease in the HBV prevalence appears to be related to the universal newborn vaccination program that started in 1998 and the adult vaccination campaign that started in 2006. The observed decrease resembles the drop in hepatitis B incidence and HBsAg positivity rates observed in other indigenous populations of the Circumpolar Arctic following the implementation of universal birth dose vaccination programs [3,42]. According to published data, hepatitis B vaccination coverage in the Russian Arctic in 2019 exceeded 90% in children and adults aged up to 35 years and 83.7% among people aged 36–59 years [17]. However, the detection of anti-HBc in 15% of individuals aged less than 30 years old indicates the remaining risks of exposure to HBV and maintenance of viral circulation after the implementation of childhood vaccination.

The HBV genotype and sub-genotype distribution observed in the surveyed cohort was similar to that reported earlier in Yakutya, including amongst indigenous populations [17,18]. Despite the low HBV DNA detection rates in HBsAg reactive sera (53%), presumably due to low viral load, the HBV genotype was deduced in all the samples available for further testing using the ELISA HBV serotyping assay. This assay has been previously shown to provide accurate HBV genotyping results with good concordance with sequencing data [43]. The results of the time-scaled phylogenetic analysis demonstrated that all HBV strains from this study were autochthonous, i.e., were not introduced recently, but resulted from the centuries-long circulation of the virus in the Arctic zone of Yakutia, similar to the HBV strains identified in Tuva, another region of the Russian Federation that was endemic for hepatitis B [26].

Anti-HDV antibodies were detected in approximately a third of HBsAg-positive participants. These data are consistent with prevalence estimates for the general population of Yakutia based on studies performed more than twenty years ago [16,44], indicating the wide and stable circulation of HDV in the region. 

The high rates of anti-HCV detection exceeding 5% demonstrate a high level of HCV endemicity in the population of Momsky district of Yakutia, with individuals aged 40 years and older having the highest risk of being infected with HCV. Moreover, our study demonstrated an increase in HCV prevalence compared to the 2% anti-HCV average prevalence rate reported in the Arctic zone of Yakutia 20 years ago [16,28]. However, an earlier seroprevalence study has shown even higher rates of anti-HCV positivity rate (7.6%) in another district of the Arctic zone of Yakutia [16].

The time-scaled phylogenetic analysis of HCV sequences isolated in our study suggests the relatively recent history of the HCV epidemic in the indigenous populations of Yakutia, with the introduction of HCV strains of subtypes 1b, 2a and 2k/1b occurring in the 1980s and 1990s. Moreover, the sequence analysis revealed a series of HCV 1b transmission events in inhabitants of a particular settlement. Taken together, these data strongly indicate the need for the implementation of universal HCV screening in the indigenous populations of the Arctic zone of Yakutia, together with HCV prevention services. As the knowledge of the participant’s infection status was not the exclusion criterion, we were able to assess the proportion of infected persons who were unaware of their infection status. Only 11 (45.8%) of 24 participants who tested positive for active HBV or HCV infection were presumably aware of their infection, as they had relevant medical records. Notably, none of them received antiviral treatment.

Our data on the viral hepatitis burden in the indigenous population of the Arctic zone of Yakutia suggest the urgent need for the development and implementation of HBV and HCV testing and therapy programs in the adult population of the region to meet the WHO elimination goals. Besides measures necessary to expand coverage with testing and treatment services, an awareness campaign among the population is needed aimed at the prevention of person-to-person transmission through household or sexual contacts.

There are several limitations of our study. First, only adult individuals were included in the survey. Thus, the data on the epidemiology of viral hepatitis in minors remain missing. Second, only one district of the Arctic zone was covered with the serosurvey, making it impossible to extrapolate prevalence data for the whole region. Third, the small number of HBV and HCV sequences isolated in this study definitely does not represent the whole genetic heterogeneity of these viruses in the Arctic zone of Yakutia. However, this is the first study that has addressed the burden of viral hepatitis in the Arctic zone within the last 20 years; it has covered almost 10% of the district population, which constitutes about 0.6% of the total population of the Arctic zone of Yakutia [14], making the study representative for the circumpolar territory of the region. The results obtained provide an understanding of current trends in the epidemiology of viral hepatitis in the Arctic zone of Yakutia.

## 5. Conclusions

Our data demonstrated the high prevalence of hepatitis B, D and C in indigenous populations inhabiting the Momsky district in the Arctic zone of Yakutia. The hepatitis B vaccination has significantly reduced the HBV infection rates in individuals aged less than 30 years old. However, the prevalence of undiagnosed HBV and HDV infections in the older population is still high. The widespread prevalence of HBV/HDV coinfection in the studied population indicates the need to expand HDV screening, improve linkage to care and treatment, and introduce measures to prevent superinfection among HbsAg-positive individuals including, but not limited to awareness campaigns among patients and healthcare providers. The high HCV detection rates, together with data from time-scaled phylogenetic analysis, indicate that the recent hepatitis C epidemic started at the end of the 20th century in the native populations of the Arctic zone of Yakutia. Taken together, these data point out the urgent need for expanded screening and care programs in this population. The HAV seropositivity in the adult population of the studied region exceeds 50%, indicating an intermediate level of endemicity. Further studies of HAV seroprevalence in children are needed to understand the possible transition in endemicity level and the necessity of an HAV vaccination program in toddlers. Anti-HEV detection rates suggest that HEV may also be the infection relevant for the Arctic zone and should be considered in the diagnosis of acute hepatitis cases.

## Figures and Tables

**Figure 1 microorganisms-12-00464-f001:**
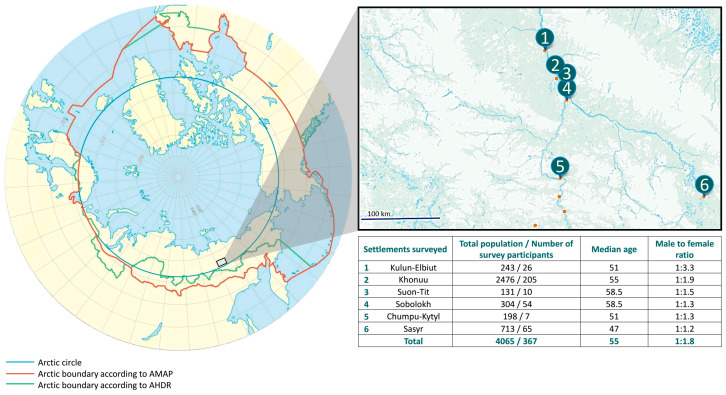
The study region (in color) shown on a map alongside the location of surveyed settlements and numbers of participants in each settlement. AMAP—Arctic Monitoring and Assessment Programme; AHDR—Arctic Human Development Report.

**Figure 2 microorganisms-12-00464-f002:**
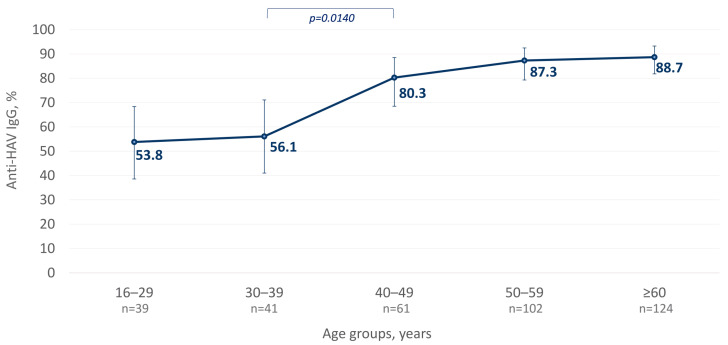
Age-specific anti-HAV IgG detection rates among indigenous populations of the Arctic zone of Yakutia; 95% CI values are shown with vertical bars. Significant differences between the age cohorts are highlighted in colored brackets with the *p*-value indicated below (Fisher’s exact test).

**Figure 3 microorganisms-12-00464-f003:**
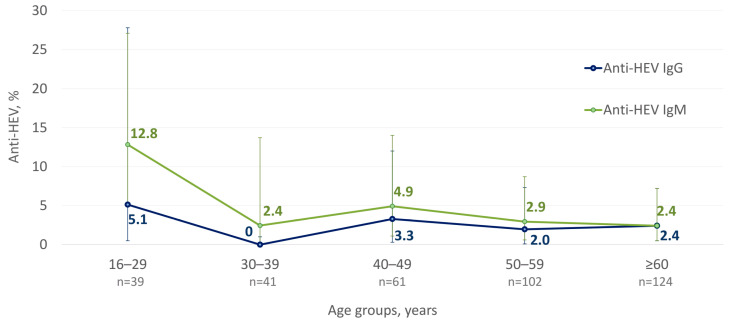
Age-specific anti-HEV IgM and IgG detection rates among indigenous populations of the Arctic zone of Yakutia; 95% CI values are shown with vertical bars.

**Figure 4 microorganisms-12-00464-f004:**
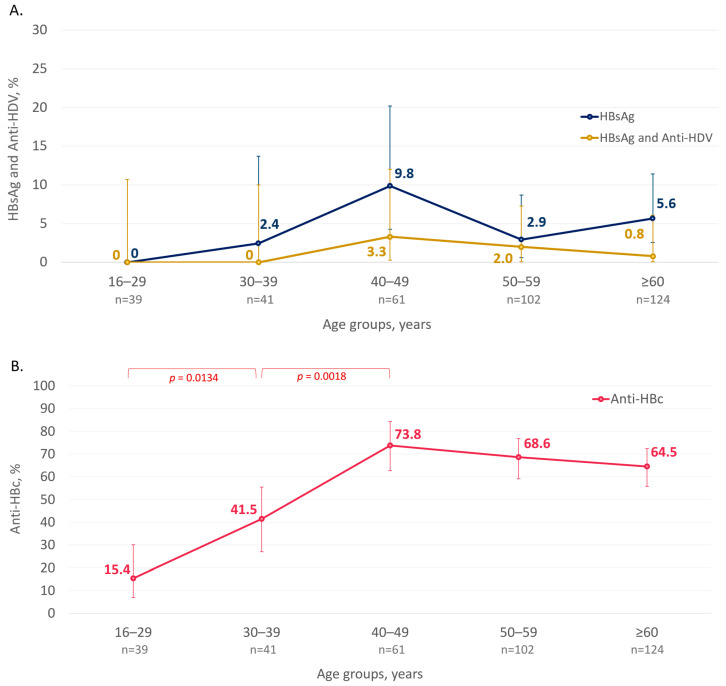
Age-specific detection rates of HBsAg and HBsAg/Anti-HDV (**A**) and Anti-HBc (**B**) among the indigenous populations of the Arctic zone of Yakutia. The 95% CI values are shown with vertical bars. Significant differences between the age cohorts are highlighted in colored brackets with the *p*-value indicated below (Fisher’s exact test).

**Figure 5 microorganisms-12-00464-f005:**
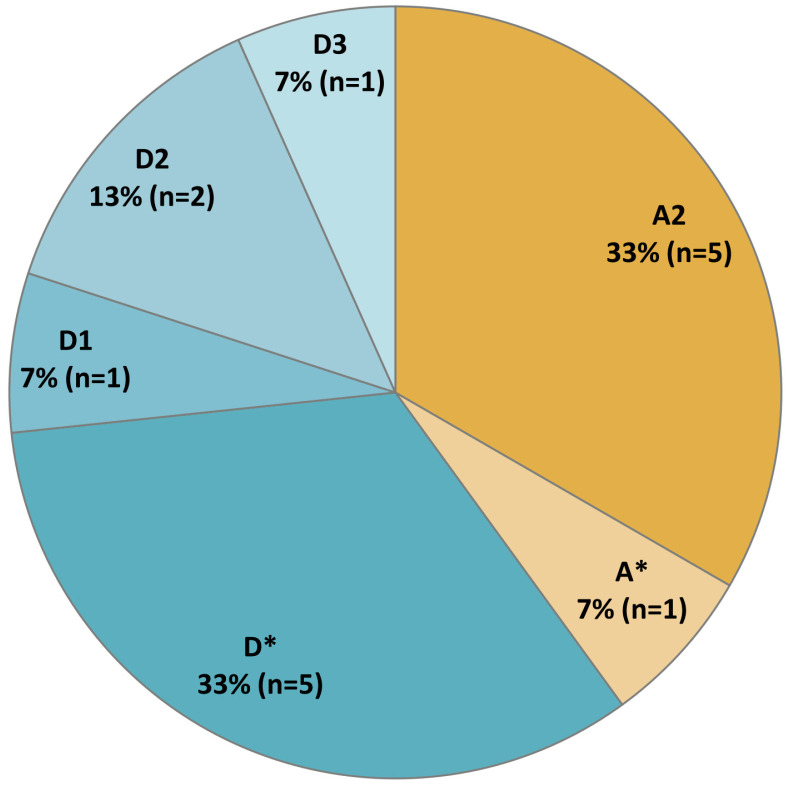
The distribution of HBV genotypes in HbsAg-positive participants. HBV genotypes identified using ELISA assay without further discrimination between sub-genotypes are indicated with an asterisk. HBV sub-genotypes are identified based on the phylogenetic analysis of the nucleotide sequences.

**Figure 6 microorganisms-12-00464-f006:**
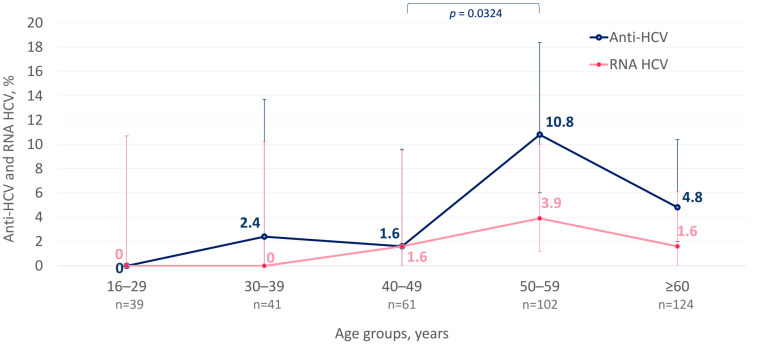
Age-specific detection rates of anti-HCV and HCV RNA among indigenous populations of the Arctic zone of Yakutia. The 95% CI values are shown with vertical bars. Significant differences between the age cohorts are highlighted in colored brackets with the *p*-value indicated below (Fisher’s exact test).

**Figure 7 microorganisms-12-00464-f007:**
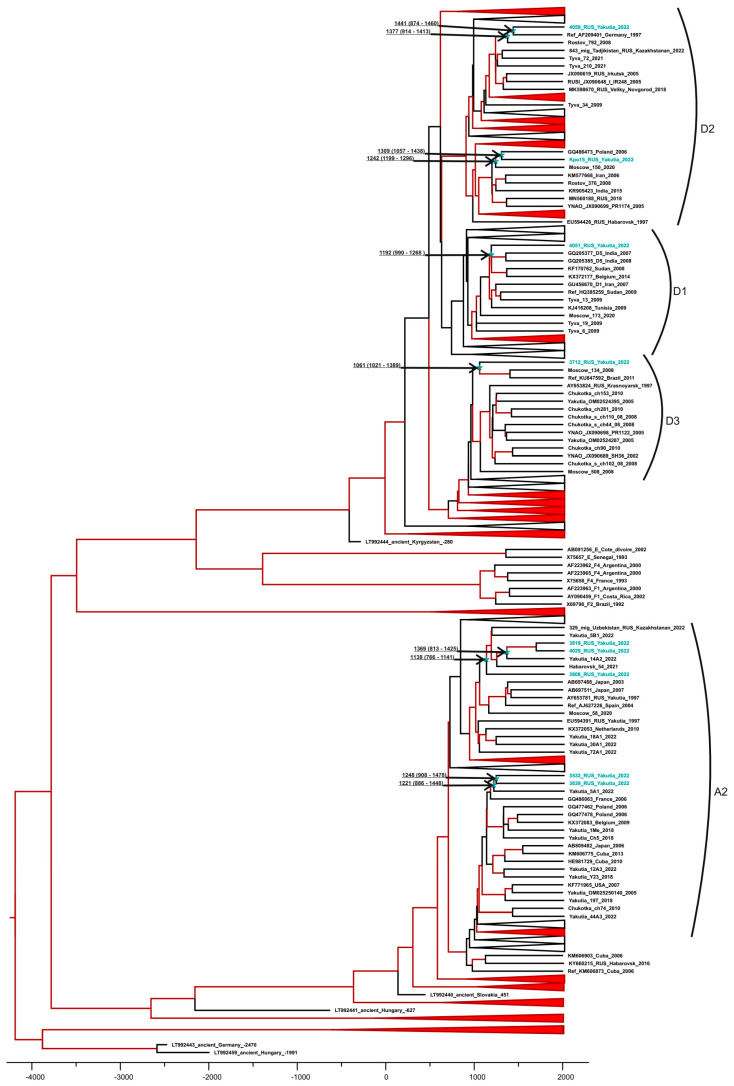
Bayesian phylogenetic tree based on tree based on HBV S-gene partial sequences (676 nt, nucleotide positions 149–824 by reference sequence NC_003977.2). The tree clusters not related to sequences from this study are compressed to ensure the visibility of the tree. Reference sequences are shown with GenBank database, the HBV genotype, the country and the year of isolation, as well as region or city if sequences are from Russia. Sequences from this study are shown in turquoise. Tree branches indicated in red have a posterior probability > 90%. In tree nodes related to sequences from this study (indicated with an asterisk), the time of the MRCA is shown with 95% HPD. The X-axis displays chronological time in years.

**Figure 8 microorganisms-12-00464-f008:**
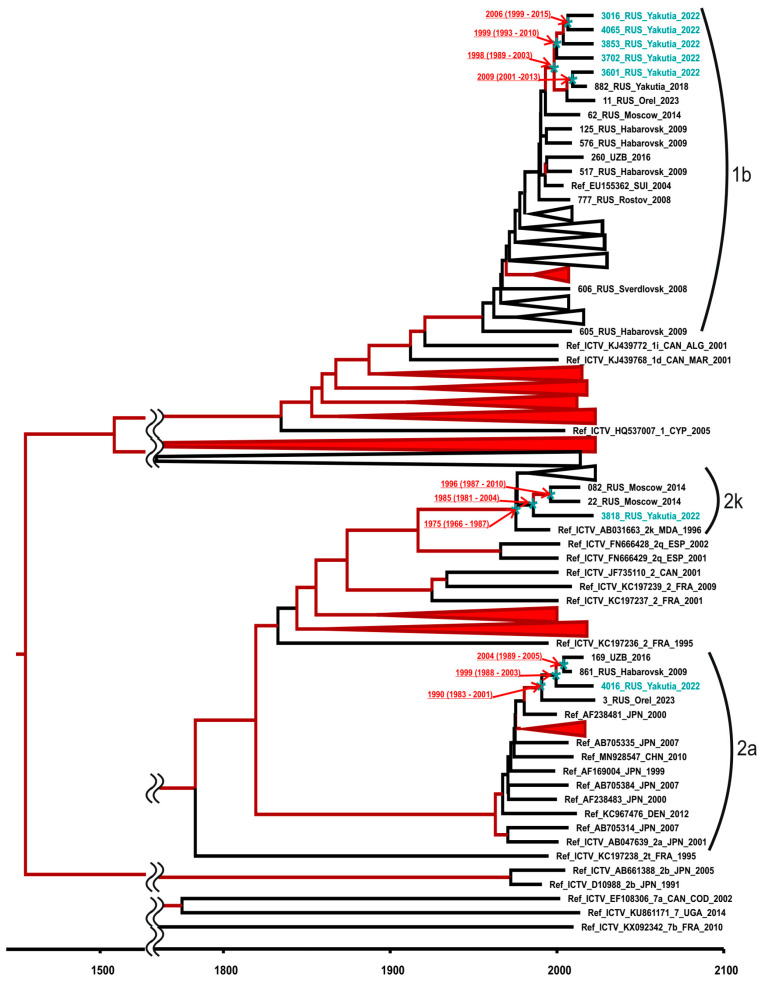
Bayesian phylogenetic tree based on 942 nt HCV sequences encoding core and E1 fragment (nucleotide positions 293–1234 according to H77 reference strain, GenBank accession number AF011753). Reference sequences are shown with GenBank accession number, the HCV genotype, the country and the year of isolation, as well as the region or city if sequences are from Russia. Sequences from this study are shown in turquoise. Tree branches indicated in red have posterior probability >90%. In tree nodes related to sequences from this study (indicated with an asterisk), the estimated MRCA time is shown with 95% HPD. Major cluster roots were cut off for the visibility of the modern clusters. The X-axis displays chronological time in years.

## Data Availability

The data presented in this study are available in this article and its Appendix A.

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
