# Peer review of "Epidemiology of Viral Hepatitis in the Indigenous Populations of the Arctic Zone of the Republic of Sakha (Yakutia)"

_microorganisms, 2024, doi:10.3390/microorganisms12030464_

Round 1

Reviewer 1 Report (Previous Reviewer 1)

Comments and Suggestions for Authors

This is a huge paper on hepatitis viruses in indigenous inhabitants of the Arctic zone living in six settlements located in the Momsky district of Yakutia. It was necessary for the authors to organize several aspects related to collection and logistics to reach these results. What has already been written is sufficient for publication. Reviewing the data on HDV, it was unclear whether these cases were HDVPCR positive and whether they were sequenced and genotyped and which HBV genotypes were ELISA or PCR positive for HDV. If they had HDV sequences, these should go to Genbank too. Finally, add something about HDV in the conclusions. These data would further enrich this important article.

Author Response

Reviewer 2 Report (New Reviewer)

Comments and Suggestions for Authors

This manuscript is relevant for the field since the presented results provide additional information on the prevalence of viral hepatitis in the indigenous populations of the Arctic zone. The manuscript is well written and I personally really like the part where the authors listed the limitations of the study.  However, I do suggest certain things, which need attention, improvement and clarification to support and strengthen the overall impact of the article.

Major points for attention:

Abstract: The abstract should be a total of about 200 words maximum, please revise.

Keywords: Please reconsider the choice of the keywords in order to increase the paper’s searchability. Use Keywords to identify the object, problem and method of study. Keywords should list the main topic of the paper for indexing purposes, so they should not be too general.

Introduction:

Line 60-61: Please indicate incidence rates of HBV and HCV.

Materials and Methods:

Authors should indicate the storing conditions of prepared RNA isolates (indicate whether they were processed immediately after extraction or stored).

Lines 152/153 and 166/167 are more suitable for chapter Results.

Results:

Authors should indicate from which article they adopted the proposed phylogeny and reference sequences (besides similar sequences obtained by BLAST), this should be indicated in the text and also in the both Figures. 

Discussion: /

Conclusions:/

Minor points for attention:         

Abbreviations e.g., HAV, HBV, HCV, HDV, HEV are not used properly. They should be spelled out at their first appearance and used as abbreviated later, please, revise throughout the manuscript.

References should be cited according Microorganisms citing style (in Abbreviated Journal Name the “full stop” should be inserted after every word abbreviation). Please take a look at the current issue of Microorganisms.

Author Response

This manuscript is a resubmission of an earlier submission. The following is a list of the peer review reports and author responses from that submission.

Round 1

Reviewer 1 Report

Comments and Suggestions for Authors

The study "Epidemiology of viral hepatitis in the indigenous population of the Arctic zone of the Republic of Sakha (Yakutia)" is very interesting as it deals with the five hepatotropic viruses and analyze serological and molecular markers together with sequencing data, genotyping and bayesian analysis from sequencing data.

This kind of study is very relevant to analyze the origins and the future of viral hepatitis around the world, involving remote populations that probably took an important role both in the surging and initial viral population as well as when we look to eliminate them as a health problem.

HAV and HBV distributions reflect the importance of viral vaccination in controlling these diseases. HEV RNA were not detected, probably due to the very short viremic period. HEV antibodies were present in 12.8% individuals with 16 to 29 years old. It is interesting that HCV was more frequent in the ages 50 to 59.

Data on TMRCA for HBV and HCV in Yakhutia are very interesting and consistent with previous reports.

Some comments:

It would be interesting to know which as the proteins that are detected using the anti-HEV kits (DS-EIA-ANTI-HEV-G and DS-EIA-ANTI-HEV-M, 112

Diagnostic Systems, Nizhniy Novgorod, Russia)

Some corrections:

1)HBV sequences from this study have been deposited in Genbank under accession numbers …–… (submission ID=2781739, 2781778).

…under accession numbers 2781739 – 2781778.

 2) HCV sequences from this study have been deposited in GenBank under accession numbers …– … (submission ID=2781904, 2781955).

…under accession numbers 2781904 - 2781955.

3) All testing performed with commercial ELISA and PCR assays were performed (was done) according the instructions of the manufacturers of the respective kits

Reviewer 2 Report

Comments and Suggestions for Authors

The manuscript of Kichatova et al described the prevalence of viral hepatitis in an Arctic regions of Russia. There is few data available in Russia about the prevalence of viral hepatitis and the paper is well written.

Hoewever, the serological tests used to conduct these study are manufactured in Russia. We did not known the performances of these tests (sensitivity and the specificity) and they are not used in others countries.

HBV sequencing genotyping is mix with HBV serotyping which do not give the same analytical details. Papers on HBV do not usually mixed genotyping and serotyping.

The paper lacks of scientific strenght for publication in this journal.

Reviewer 3 Report

Comments and Suggestions for Authors

The authors have assessed the prevalence of hepatitis viruses and herd immunity among the Native people in the Arctic zone of Yakutia. The work sheds light on an area that has been notably underrepresented in studies, and your findings are invaluable in understanding the health landscape regarding viral hepatitis in this region. However, I believe the manuscript could be further improved.

Abstract

1.     Line 18 – Authors state that the average HAV seroprevalence was above 50%. It is important to mention the exact prevalence in the abstract.

2.     Were there any significant sex/gender differences in the prevalence of the different viral hepatitis? That should be reported.

Introduction

1.       Line 38: Authors should kindly check the grammar of the sentence to improve clarity.

2.       The description of the study setting is excessively long for the introduction. Could have easily been in the methods under study setting. The long descript still lacks important information on the health system, universal health coverage availability, and how the health system promotes prevention, screening, and treatment of viral hepatitis.

3.       The introduction implies that the indigenous people in this area are labeled as "hard-to-reach and drastically affected." Is the difficulty in accessing this population greater in comparison to migrants within the region, or is this assessment is in regards to other regions?

Methods

1.     Very important. How was the random selection of study participants done? Was the selected population representative of the overall population of Yakutia?

2.     Did the study include participants with know viral hepatitis diagnosis? Was this an exclusion criterion or not? If it wasn’t an exclusion criterion, what are the characteristics of these individuals? Were they receiving treatment?

3.     Line 91-93: these are results and should be in the appropriate section.

4.     Line 88: Which demographic data was collected in the study? Kindly state clearly.

5.     Line 89: The male to female ratio was 1:1.8. – Again, these are results and should be in the appropriate section.

6.     Line 91-93: According to the questionnaire data, the majority of participants self-identified as Yakuts (49.6%), Evens (38.7%), Evenks (10.1%), Dogans (0.5%), Russians (0.5%), and 0.5% of participants did not answer this question. – How did the authors define natives/indigenous people?

7.     Ethics issues should be addressed once. Participants informed consent forms+ethical clearance. It is difficult to understand why these are under “2.1. Serum samples” in methods.

8.      Statistical analysis should be improved. We should be able to replicate the results looking at this session but that is currently not the case.

Results

1.     Ideally, table 1 should give us a summary of all study participants according to all the variables collected in the survey which is missing. This also helps in the generalizability of findings in other settings.

2.     Line 205: 3.1. HAV and HEV testing results – Authors should consider beginning the various subsections with the identified prevalence of the viral hepatitis type in question.

3.     What are the characteristics of participants who were HDV positive? How different are they from those with HBV positivity without HDV?

4.     Line 281: Improve the grammar for clarity please.

Discussion

1.     Could authors provide some context on access to health services including testing and vaccination for the conditions discussed? We know the mentioned the doses given at birth since 1998 but what about those who were born before then? How is access like for them? Are there any specific public health programs in place to improve access?

2.     Even though only one district was considered in the arctic region, could authors tell if the sociodemographic makeup.

3.     The discussion lacks what the findings mean for public health and the recommendation on meeting the WHO targets. Elaborating further on the implications of your results within the context of public health policies could add depth to the discussion section.

Comments on the Quality of English Language

The study could benefit from English editing services to enhance the clarity of some sections. 
